# Specific Changes in Morphology and Dynamics of Plant Mitochondria under Abiotic Stress

**Hui Tang and Hongliang Zhu *** 

The College of Food Science and Nutritional Engineering, China Agricultural University, Beijing 100083, China
* Correspondence: hlzhu@cau.edu.cn

**Abstract:** As the global climate continues to warm and the greenhouse effect intensifies, plants are facing various abiotic stresses during their growth and development. In response to changes in natural environment, plant mitochondria regulate their functions through morphological and dynamic changes. Mitochondria are highly dynamic organelles with the ability to continuously cleavage and fuse, regulating dynamic homeostatic processes in response to the needs of organism growth and the changes in external environmental conditions. In this review, we introduced the structure of the outer and inner mitochondrial membrane and discussed the relevant factors that influence the morphological changes in mitochondria, including proteins and lipids. The morphological and dynamic changes in mitochondria under various abiotic stresses were also revisited. This study aims to discuss a series of changes in plant mitochondrial ultrastructure under abiotic stress. It is very important that we analyze the association between plant mitochondrial functions and morphological and dynamic changes under stress to maintain mitochondrial homeostasis and improve plant stress resistance. It also provides a new idea for plant modification and genetic breeding under the dramatic change in global natural environment.

**Keywords:** mitochondria; morphological and dynamic changes; mitochondrial ultrastructure; cristae; abiotic stress; oxidative stress

## 1. Introduction

Plant mitochondria are two membranous organelles of endosymbiotic origin with their own genetic information. As the energy factory of cells, the main functions of plant mitochondria are producing ATP through tricarboxylic acid cycle and oxidative phosphorylation, releasing energy and producing various metabolic products to participate in programmed cell death (PCD), oxidative stress, and other key cellular processes. All of them are crucial for the maintenance of homeostasis in eukaryotes and the growth of organisms.

In higher plants, mitochondria are usually spherical, sausage-shaped, linear, or network shaped, and their morphology is highly variable [1]. They are usually distributed in the cytoplasm [2] with diameters ranging from 0.2 to 1.5 μm [3]. Mitochondria dynamic processes can be divided into fusion and fission. They can fuse and connect with each other to form network-like structures or split to form dispersed individuals. The movement of mitochondria depends on cytoskeleton, which is composed of actin filaments, intermediate filaments, and microtubules [4]. They move rapidly along tubulin and actin filaments in cellular mitochondria of which processes are regulated by protein kinases [5,6]. Based on their morphological and dynamic characteristics, mitochondria can adjust their shape, number, and orientation according to the developmental needs of plant cells [7,8]. Such morphological and dynamic changes will eventually lead to changes in mitochondrial functions, which gives them the ability to adapt to changes in the external environment.

With the global warming, plants on land are constantly affected by various adverse or even negative environmental conditions. Among them, abiotic stress (e.g., high temperature, low temperature, salt, drought, ozone, UV radiation, etc.) has adverse effects on plant

growth and productivity [9]. Plants respond to abiotic stress in various ways. One of the most obvious features is the induction of excessive cellular production of reactive oxygen species (ROS). Ultimately, it may lead to PCD [10].

In plant cells, the molecular and physiological responses of mitochondria to stress are well known. As a powerhouse organelle in eukaryotes, mitochondria produce large amounts of ATP, which can provide 95% of the energy required for life activities. It is the main site for regulating apoptosis and producing ROS [11]. Therefore, plant mitochondria play a key role in the process of responding to abiotic stress. Under stress, plant mitochondria sense metabolic changes, such as pH, energy status, and ROS, and respond to them by inducing permeability transition pores (PTP) and releasing cytochrome c. When stress persists and intracellular ROS exceed the regulatory threshold of mitochondria, they will break down and lose functions, triggering a cascade of PCD in cells [12–16], which ultimately leads to the obstruction of plant growth and development. Experiments have confirmed that mitochondria produce a large number of ROS and the loss of outer membrane potential under various stimuli, which were considered to be the early product of PCD in *Arabidopsis thaliana* [17]. Therefore, it is necessary to focus on the morphological and dynamic changes in mitochondria induced by abiotic stress to understand their functions. In this review, we introduced the ultrastructure of plant mitochondria in plants, highlighting the specific morphological and dynamic changes in mitochondria under abiotic stress, with the aim of discussing possible mechanisms by which the pattern of changes in plant ultrastructure under extreme climates affect their growth and development.

## 2. Morphological and Dynamic Changes in Mitochondria

### 2.1. Ultrastructure of Mitochondria

Mitochondria are two membrane-closed organelles whose morphological characteristics include four functional regions from the outside to the inside: outer membrane (OM), intermembrane space (IMS), inner membrane (IM), and mitochondrial matrix (Figure 1). Mitochondrial outer membrane separates the mitochondria from the cytoplasmic matrix. Their IM can be divided according to its different functions into inner boundary membrane (IBM), which is close to and parallel to OM and cristae membrane (CM), which bulges toward the matrix [18,19]. Notably, CM is not a simple extension and folding of IM. Instead, it is connected to IBM through a narrow tubular cavity called cristae junctions (CJs) [19,20], which have been described as the third compartment of mitochondria [21]. IBM and CM have different topologies and protein compositions, where IBM is located next to OM and possesses the mechanism by which most proteins enter inner mitochondria. While CM is the main site of oxidative phosphorylation and contains the complex of respiratory chain as well as F1Fo-ATP synthase [22,23]. An increasing number of studies has demonstrated that the shape of CJs is related to the process of mitochondrial fusion and fission [24,25], especially to PCD [22,26,27].

### 2.2. Structure of Cristae

In plants, although significant progress has been made in identifying proteins involved in mitochondrial morphology, little is known about the protein complexes that control CJs biogenesis. In yeast and mammals, the large GTPase optic atrophy 1 (OPA1), the mitochondrial contact site and cristae organizing system (MICOS), and the IM protein dimer F1Fo-ATP synthase are involved in the regulation of cristae morphology [22,27,28]. Mammalian OPA1 (Mgm1 in yeast) is involved in remodeling and fusion of the mitochondrial inner membrane [29–31]. In addition, MICOS is an evolutionarily conserved large heterooligomeric protein complex of IM, mainly located in CJs [32–35]. MICOS is composed of two different subcomplexes, with MIC10 and MIC60 (also known as mitogen) as the core components [18], which maintain the stability of normal mitochondrial endometrium cristae and are crucial for the formation of cristae [33,36,37]. The overexpression of *Mic60* and *Mic10* leads to the bifurcation of cristae [38] and the strong expansion and deformation

of CM and CJs [39]. In plants, according to phylogenetic analysis of eukaryotes, only MIC60 and MIC10 are conserved [40,41].

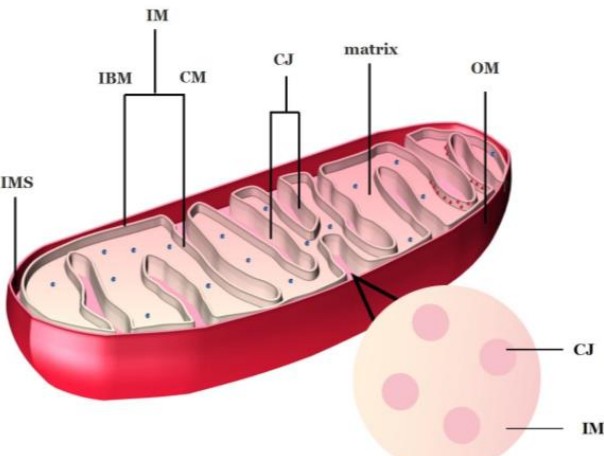

**Figure 1.** Mitochondrial structure of plants. IMS: intermembrane space; IM: inner membrane; CM: cristae membrane; CJ: cristae junctions; IBM: inner boundary membrane; OM: outer membrane. The blue spots in the matrix are ribosomes; the red spots in the CM are mitochondrial respiratory chain complexes. Image credit: Hui Tang.

MICOS, as a large multi-subunit complex involved in mitochondrial biogenesis and stability, is closely related to the maintenance of mitochondrial structure. The absence of several MICOS subunits leads to intense changes in mitochondrial cristae morphology, including the loss of cristae and the formation of cristae stack (which appear as stacks within the mitochondrial matrix) [19,42,43]. Moreover, lipid exchange is also necessary to maintain the integrity of the mitochondrial membrane. An Arabidopsis study has shown that MIC60 interacts with the mitochondrial outer membrane transportase (TOM) through the TOM 40-kD subunit (TOM40) to form part of the mitochondrial transmembrane lipoprotein (MTL) complex, which influences the mitochondrial lipid transport [44]. In addition, Mic60, together with the mitochondrial outer membrane protein DGD1 suppressor 1 (DGS1), forms multi-subunit complexes in *Arabidopsis* [45]. The loss of DGS1 resulted in the whole plant physiology being affected, namely it altered the stability and protease accessibility of this complex and altered the lipid content and composition of the mitochondria, particularly causing changes in the abundance and size of cristae in the morphology [45].

The energy conversion of the mitochondrial respiratory chain is achieved by electrochemical proton gradient inside and outside the mitochondrial inner membrane. This potential gradient is then utilized by the F1F0-ATP synthase to produce ATP. The F1Fo-ATP synthase, one of the mitochondrial electron transport chains (ETC) complexes, i.e., complex V, exists as a dimer and is found along the most tightly curved regions of the cristae ridge or around narrow tubular fractures [22,46,47]. In plants, ATP synthase accounts for about 8.44% of the volume of the mitochondrial inner membrane, which is assembled together with complex I~IV on mitochondrial cristae to form multiple and bulky ETC complexes, which jointly affect the mitochondrial inner membrane structure [48]. It has been proved that the dimer state of F1Fo-ATP synthase can affect the structure of cristae [48] and the deletion of ATP synthase subunits e and g leads to "onion-like" structures of cristae [28,47,49]. Strauss and Hofhaus [50] have found that the control of ATP synthase on cristae morphology and the ATP synthase would exert a strong local curvature on the membrane, resulting in the formation of cristae so that protons would gather in the cristae. At this time, mitochondrial cristae act as a proton trap, and ATP synthase can achieve more efficient ATP synthesis by changing the mitochondrial inner membrane morphology driven by strong proton power [50].

Mitochondrial membrane structure and cristae shape are also affected by some proteins that affect membrane stability. Membrane-anchored ATP-dependent metalloproteinases

called FtsH4 or AAA proteases are believed to be key enzymes for the quality control of membrane proteins in mitochondria and chloroplasts [51]. The absence of AtFtsH4 changed the leaf morphology of the rosette at the later stage of development, and the number of mitochondrial cristae decreased significantly in the micromorphology of organelles during the short-day cycle [51].

At present, the molecular mechanism of the morphological change in plant mitochondrial cristae is not perfect, as a result of the structure of plant cells is more complex. In order to fill this gap, maybe we can refer to the above reports to promote the study of plant mitochondrial morphology.

### 2.3. Dynamic Changes in Mitochondria in Plants

2.3.1. Proteins Associated with Mitochondrial Fusion

The dynamic processes of mitochondria depend on the proteins involved in the regulation of mitochondrial formation, fusion, and fission.

In mammals and yeast, the dynamin-related protein 1 DLP1/Drp1(the yeast homologous Dnm1p), which is a superfamily member of large guanosine triphosphatases (GTPases), mediates the dynamic process of mitochondrial division [52]. Located mainly in the cytoplasm, DLP1/Drp1 has a highly conserved NH2-terminal GTPase domain at the N-terminal, followed by an intermediate domain, and a GTPase effect-domain (GED) with mitochondrial targeting at the C-terminal [53]. Mitochondrial fission requires recruitment of Drp1 to the mitochondrial surface and activation of its GTP-dependent fission function. In mammals, the primary receptors that recruit Drp1 to facilitate mitochondrial fission include mitochondrial fission 1 protein (Fis1), mitochondrial fission factor (Mff), and mitochondrial dynamics proteins of 49 and 51 kDa (MiD49 and MiD51, respectively). They drive DRP1-mediated division by interacting [54] or recruiting different subgroups of Drp1 [55]. There are mitochondrial fission factors in *Arabidopsis*, dynamin-related proteins DRP3A and DRP2B (ADL2a and ADL2b, the functional orthologs of Dnm1p and DLP1/Drp1), thought to be key factors in both mitochondrial and peroxisomal fission. In the double mutant, *drp3a/drp3b*, the mitochondria are connected to each other, resulting in massive elongation [56]. They are functionally redundant in mitochondrial fission, but the frequency of mitochondrial fission in Arabidopsis depends on the total abundance of DRP3A and DRP3B [56]. In *Arabidopsis thaliana*, there are two closely related Fis1 homologues (i.e., FIS1A and FIS1B) that have been reported to target mitochondria, peroxisome, and chloroplast, which have been shown to promote mitochondrial fission [57,58]. From now on, there is no other homologue of the mitochondrial fission factor (Mdv1p/Caf4p, Mff or MiD49/51) in *Arabidopsis thaliana*. However, there are Arabidopsis-specific fission factors–elongated mitochondria 1 (ELM1) and peroxisome and mitochondrial division factors (PMDs). ELM1 localizes to the outer surface of mitochondria and mediates mitochondrial fission by recruiting DRP3A; moreover, mitochondria in *elm1* mutant showed a slender shape [59]. However, *elm1* mutant has residual mitochondrial fission activity [59], which may be caused by a completely independent mitochondrial fission system. It has been reported that elongated mitochondria had been observed in *pmd1* mutants; however, PMD1 did not physically interact with DRP3 or FIS1 [60]. It suggests that PMD1 may promote mitochondrial proliferation in an independent manner from DRP3/FIS1.

2.3.2. Proteins Associated with Mitochondrial Fission

Mitochondrial fusion is also regulated by large GTPases, which requires three steps: the connection of two mitochondria, fusion of the OM, and fusion of IM [61,62].

Mitofusin (Mfn)1 and Mfn2, (homolog Fzo1 in yeast) are involved in mitochondrial tethering and OM fusion. In the absence of Mfn1 or Mfn2 cells, the imbalance of the fusion and fission events leads to mitochondrial fragmentation, leading to severe mitochondrial and cellular dysfunction [63,64]. Mfns structurally contain the C-terminal GTPases region, middle region (MD), transmembrane region (TM, including TM1 and TM2), and the GTPase effector domain (GED). MD and GED are two predicted heptad repeat domains (HR1 and

HR2). The sequence composed of eleven amino acids in the middle of TM1 and TM2 region was identified as mitochondrial targeting sequence MTS. It facilitates anchoring and targeting of these proteins to the mitochondrial membrane [65]. The hetero-type complex formed by Mfn1-Mfn2 or the single homologous Mfn1 or Mfn2 complex are important regulators of mitochondrial outer membrane attachment and fusion [64,66].

Human optic atrophy-1 (OPA1) (homologue Mgm1 in yeast) is mitochondrial localization protein, which is a member of the dynamin family. Thus, it has the characteristic structure of the dynamin family, including N-terminal mitochondrial targeting signals (MTS) and transmembrane domains, GTPase domain, a central domain, and a GED at the C-terminal. When the OPA1 imports into mitochondria, its matrix-targeting signal is removed, and as an L-isoform tightly bound to or embedded in IM, the rest resides in the mitochondrial intermembrane space [67,68]. In general, OPA1 mediates IM fusion and maintains the cristae structure. When OPA1 is deactivated, it can cause mitochondria to fragment [69].

From now on, no specific fusion factor has been found in plants. Mitochondria often presents as highly fragmented and suggests that plants with the dynamical process of mitochondria are mainly composed of fission. However, fusion phenomenon does occur. For example: mitochondria in lower plants, such as the unicellular freshwater alga *Micrasterias denticulata*, are globular and exist independently of each other in the cytosol at room temperature. As the temperature decreases, mitochondria begin to aggregate and fuse with each other. When subjected to extracellular freezing stress at $-2\,^\circ$C, *Micrasterias* mitochondria are aggregated into local networks, and their OMs are connected or fused with each other and finally aggregated into a large mitochondrial network [70]. In higher plants, Arimura [71] demonstrated the fusion of mitochondria through an interesting phenomenon. They labeled mitochondria in onion epidermal cells with mitochondria-targeted, photoconvertible fluorescent protein Kaede and then used light processing to turn some of the mitochondria within a cell from green to red. Finally, they found the appearance of yellow mitochondria.

### 2.3.3. Fusion and Fission of Cardiolipin with Mitochondria

The composition of the mitochondrial membrane is also critical to the dynamic process of mitochondrial fusion and fission. Mitochondrial membrane is composed of phosphatidic acid (PA), phosphatidylserine (PS), cardiolipin (CL), phosphatidylethanolamine (PE), phosphatidylglycerol (PG), phosphatidylcholine (PC), and phosphatidylinositol (PI), in which PC and PE are the main components. One of the potential mechanisms by which lipids affect mitochondrial morphology may be due to their ability to recruit and/or activate proteins that mediate the process of fission and fusion. For example, as a precursor for glycerol lipid synthesis, which is produced by Mt-GPAT functions, LPA produced on OM may stimulate GTPase activity of Mfns and enhance mitochondrial fusion during OM fusion [72]. Similar stories were reported in CL. CL is unique in mitochondria and significantly enriched in IM, accounting for 10–20% of total phospholipids [73], and even up to 25% at the contact site between IM and OM [74]. In yeast, CL binds to the soluble form of Mgm1 in order to stimulate its GTPase activity and affect the MGM1-mediated IM fusion process [75]. Additionally, its cardiolipin unsaturation level is the key to mitochondrial functions and IM integrity [76]. In the mitochondria of *Arabidopsis thaliana*, cardiolipin content in *ftsh4* mutant leads to the deregulation of mitochondrial dynamics and causes perturbations within the OXPHOS complexes, which generates more reactive oxygen species and less ATP [77]. In addition, CL is also involved in the remodeling of the highly folded cristae of mitochondria [78] and affects mitochondria-mediated apoptosis and other mitochondrial functional processes. It is reported that CL is a key determinant for mt-DNA stability and segregation during mitochondrial stress [79]. Thus, these results further demonstrate the importance of CL not only for mitochondrial functions but also for the health of the organism as a whole.

### 3. Mitophagy

During mitochondrial respiration, the electron transport chain in IM generates membrane potential and ATP, which is the main site for the ROS production. When mitochondria are subjected to oxidative stress, mitochondrial osmotic conversion, mt-DNA damage, and excessive ROS are produced to disturb cell homeostasis [80]. Therefore, a quality control system is needed to clean up cellular damaged mitochondria to prevent mitochondrial dysfunction and minimize the spread of toxic substances. Mitophagy is considered to be the central mechanism for mitochondrial quality and quantity control, which is an evolutionarily conserved process [81]. Damaged mitochondria are specifically wrapped into the autophagosome and fused with the lysosome, thus completing the degradation of mitochondria and maintaining the stability of the intracellular environment.

The role of mitophagy has been extensively studied in yeast and mammals. In yeast, the mitophagy mediated by Atg32, which is an autophagy related gene in yeast, is used to optimize the quality and quantity of mitochondria [82,83]. In mammal, autophagy is also essential for controlling mitochondrial quality, as its impairment is associated with neurodegenerative and cardiovascular diseases in human [81,84]. The controlled clearance of dysfunctional mitochondria through mitophagy is an important aspect of maintaining human health.

Currently, 43 autophagy related genes (*Atg*) have been identified in fission and budding yeast [85]. Most core *Atg* genes are largely conserved in plants: direct homologues of 20 core *Atg* genes from yeast have been reported in *Arabidopsis thaliana* [86]. Additionally, they have similar functions to their yeast counterparts [87,88].

In plants, mitophagy mediated changes in mitochondrial morphology are associated with mitochondrial quality control. In the UV-B damaged *atg* mutants of *Arabidopsis thaliana*, mitochondria were highly fragmented and increased in number [89], indicating that cells are unable to clear damaged mitochondria when mitophagy was blocked. During somatogenic senescence in cells, the number of mitochondria is significantly reduced [90]. Moreover, mitochondrial fragmentation is accompanied by a decrease in mitochondrial volume prior to PCD [91]. Therefore, mitophagy may be involved in mitochondrial degradation to maintain cell homeostasis during PCD in plant cells. In another study, mitophagy reduced after it mutated fusion- and fission-associated proteins [92]. All these suggest that the process of mitochondrial morphology is closely related to mitophagy.

### 4. Interactions between Organelles Affect Mitochondrial Dynamics Processes

Cellular organelles are inextricably linked to each other. Among them, membranous organelles, with the endoplasmic reticulum (ER) at their core, form a fine and complex network of interactions that coordinate with each other to perform a series of important physiological functions.

In higher plants, the polygonal structure and motility of the endoplasmic reticulum promote mitochondrial fusion and the formation of elongated mitochondria [3]. A close association between the endoplasmic reticulum and mitochondria has been found to be important for lipid synthesis and trafficking as well as the maintenance of normal mitochondrial morphology [44,93]. In Arabidopsis, the GTPase structural domain of AtMiro2 regulates ER–mitochondria interactions, facilitating mitochondrial fusion and inhibiting motility [94]. Chloroplasts are another semi-autonomous organelle with a double membrane structure in plant cells and also have their own genome. In plants, mitochondria are usually clustered around chloroplasts [2,95], which may facilitate energy conversion between mitochondria and plastids. Mitochondria can dissipate excess reducing equivalents in chloroplasts to maintain optimal plant growth [96]. Peroxisomes are single-membrane organelles in eukaryotes with a diameter of 0.1–1 μm and do not contain their own genome [97]. In mammals, linkages are formed between mitochondria and peroxisomes [98], and in plant cells, mitochondria are spatially tightly linked to peroxisomes and chloroplasts [99], and it was found that, under light induction, peroxisomes and chloroplasts interact and recruit mitochondria to form a three-organelle complex [100]. This complex network formed

between organelles may be one of the effective ways for metabolite exchange within plant cells. However, the mechanism of this intermediate is not clear yet.

In conclusion, the functional association of mitochondria with other organelles also affects to some extent the morphological changes and kinetic processes of the mitochondria themselves.

## 5. Morphological Changes in Mitochondria in Response to Abiotic Stress

*5.1. Morphological Changes in Mitochondria under Temperature Stress*

The frequent occurrence of global climate extremes will result in a higher frequency of temperature stress in plants. It has been reported that global temperature is expected to increase by 3.2 °C compared to pre-industrial levels, exceeding the Paris Agreement global target of 1.5 °C [101]. The extreme high temperature brought about by global warming will inevitably have a negative impact on the developmental growth of plants, impairing the normal functions of organelles and causing metabolic disorders. Mitochondrial swelling and ultrastructural disorders can occur during high temperature stress [102]. In both *Melanoxylon brauna* and *Glycine max* seeds, it was difficult to identify the mitochondrial cristae structure in embryonic axis cells under high temperature stress [103,104], probably because heat stress led to membrane damage, resulting in the development of mitochondrial cristae.

Low temperature or freezing stress can also affect mitochondrial morphological changes. It has been reported that the mitochondrial structure of cells in the root system of sugarcane seedlings became blurred, broken, or disappeared, and the cristae was damaged and reduced at 4 °C [105].

Although low temperature stress will destroy the normal shapes of plant mitochondria, on some level, the plant mitochondrial dynamics process will also facilitate the adaptation of mitochondria to mild hypothermia stress. Recently, it was found that mitochondria of germinating leaves changed from a highly active, elongated state to the round one with low functional activity at different stages of spring development in *Galanthus nivalis* L., as the surrounding soil temperature transitioned from low to positive temperatures. This change did not affect the germination process [106]. Additionally, the elongated mitochondria in the early stages of germination may be for better adaptation to the low temperature of the soil surface. In another study, transient chilling treatment can induce the mitochondrial division induced by DRP3A in wild-type Arabidopsis leaves, and the mitochondrial morphology is restored after a 24 h cold treatment [107]. Such cold-induced transient fragmentation may not impair mitochondrial functions. Although mitochondrial fragmentation reduces the mitochondrial area, it increases the mitochondrial number at the same time. Notably, mitochondrial fusion in Arabidopsis cells at low temperature is not mediated by the recruitment of DRP3A by ELM1 [107]. It may be due to cold treatment increasing the affinity between DRP3A and OM or that unknown cofactors, such as other proteins or lipids, which help DRP3A locate or promote the functions of other types of mitochondrial fission. Mitochondria can also maintain their functions through fusion mechanism in plants at low temperature. For example, under cryogenic emergency conditions, mitochondria in the three algae species tended to fuse, resulting in mitochondrial elongation and the formation of interconnected networks, while no changes were observed in respiratory capacity and photosynthetic rate [70].

In summary, the morphological and dynamic changes in plant mitochondria under temperature stress, whether fusion or fission, seems to be an energy compensation mechanism to create more ATP by forming a larger number of mitochondria or increasing the working area of the electron transport chain in IM.

The substances that have a stabilizing effect on membrane components may increase the tolerance of organelles to heat stress. Spermidine, a plant polyamine, is an important plant growth regulator that is closely related to plant growth, stress response, and plant disease resistance [108]. Yang [109] has found in a study that exogenous SPD normalized the chloroplast and mitochondrial ultrastructure in lettuce seedlings during high temperature stress because it can effectively maintain the stability of chloroplast and mitochondrial

double-membrane structure in lettuce seedlings. Similar findings have been reported in studies of cucumbers and ginger [110,111].

Mitochondrial protease AtFTSH4 deficiency causes the accumulation of oxidative stress markers in Arabidopsis leaf cells under mild heat stress-induced conditions [77]. It has been demonstrated that AtFTSH4 is essential for mitochondrial membrane stability and its deletion leads to impaired mitochondrial ultrastructure [51]. Therefore, how to maintain the stability of the mitochondrial membrane under temperature stress is necessary to maintain the functions of mitochondria in plants.

*5.2. Morphological Changes in Mitochondria under Drought Stress*

In most plants, drought stress will directly affect the water homeostasis of the intracellular environment, resulting in impaired organelle functions. It can lead to changes in leaf anatomy and ultrastructure, including mitochondrial morphology [112,113], and the structural breakdown of mitochondria became more severe with the increase in drought stress [114].

Drought stress in plants inhibits photosynthesis and respiration, with chloroplasts and mitochondria bearing the brunt of the stress. It has been reported that the volume of chloroplast matrix and thylakoid in spinach leaf cells increased, while the volume of mitochondria decreased [115]. This phenomenon may be due to glucose starvation deriving from decreased photosynthetic activity and the lack of starch in the chloroplasts. In another study, researchers evaluated the changes in ultrastructure in the leaves of two types of winter wheat with different drought tolerance, under individual or combined drought and heat treatment. They found chloroplasts in the leaves of non-drought tolerant wheat were swollen, photosynthetic rate decreased, and mitochondria were swollen and vacuolized with their number increased significantly under drought stress [116].

Interestingly, it has been found that an application of exogenous 24-epbrasinolide (EBR) alleviated ultrastructural damage of chloroplast under drought stress, including swelling and thylakoid arrangement disorder, and alleviated the photosynthesis inhibition induced by water stress by increasing chlorophyll content. However, it could not restore mitochondrial cristae fracture [117]. When plants are suffering from drought, stomatal closure will be induced, which will lead to the decrease in chloroplast photosynthesis, the decrease in $CO_2$ assimilation, and the disruption of intracellular energy balance [118]. Drought stress triggers the compensatory mechanism of ATP synthesis by mitochondria in plant cells when chloroplast function is blocked [119]. The activation of the ATP production pathway under stress conditions may be in response to the higher demand for ATP under stress conditions to maintain homeostasis. Mitochondria support chloroplasts by regulating photorespiratory flux and maintaining energy supply during drought stress [120]. At present, we have not cleared the sequence and extent of plant mitochondrial and chloroplast damage in plants under drought stress, although it is probably related to the species and variety of plants [121,122]. In short, concentration on the changes in organelle ultrastructure will help us to understand a series of responses of plant mitochondria and chloroplasts under drought or other stress.

At present, researchers have found that some exogenous or endogenous substances have a positive effect on mitochondrial morphological maintenance under drought stress; for example, conjugated non-covalently spermidine (CNC-Spd) and conjugated covalently putrescine (CC-Put). Morphological integrity of mitochondria of wheat germ during development under drought stress was maintained by CNC-Spd and CC-Put [123]. Under drought dress, positively charged Spd can maintain the level of -SH group in the mitochondrial membrane by non-covalently binding with acid proteins in the mitochondrial membrane to form CNC-Spd, which enhances the antioxidant capacity of proteins, which is beneficial to the integrity of mitochondrial morphology. CC-Put may stabilize the conformation and function of proteins in mitochondrial membranes by preventing mitochondrial membrane degeneration, thus preserving the integrity of mitochondrial morphology. In mammals, endogenous $H_2S$ has been found to maintain the morphology and

functions of mitochondria, which is a regulator of energy production in mammalian cells under stress conditions and can delay cell aging by reducing oxidative stress [124–126]. In *Arabidopsis thaliana*, endogenous $H_2S$ can protect the mitochondrial ultrastructure of leaves under drought stress [127]. After the mutation and overexpression of DES1 (At5g28030), a key enzyme for $H_2S$ synthesis, the researchers observed the changes in mitochondrial ultrastructure in aging leaves under drought stress. With the occurrence of stress, the mitochondrial membrane was deformed, the mitochondria in WT and *des1* lost their internal structure, and the cristae was seriously damaged and swollen. Compared with WT and *des1*, the mitochondria in *OE-DES1* were complete and the activity level of ATPase was increased [125]. This indicates that $H_2S$ determines mitochondrial regulatory energy and is crucial for mitochondrial homeostasis, which is consistent with reports in animals [128].

In addition to the above, many compounds, such as hydrogen peroxide ($H_2O_2$) [129], abscisic acid [130,131], melatonin [132], plant growth regulators melafen and pirafen [133], triadimefon (TDM) [134], malic enzyme [135], mitochondrial alternative oxidase AOX [136], and mt-DNA binding protein WHY2 [137], play a protective role in plant drought tolerance. Atkin and Macherel [120] believed that the existence of highly conductive mitochondrial potassium channels found in plants may be the key to understanding the changes in mitochondrial matrix volume under drought conditions. This system has been reported to be related to energy dissipation and cellular stress defense mechanisms in plants [138–140] and can regulate mitochondrial volume [141]. This implies that the potential gating of potassium channels by different compounds (nucleotides, NADH, metals, proteins, etc.) may play an important role in the mitochondrial response to drought stress, which may provide some theoretical support for the study of drought resistance mechanism in plants.

*5.3. Morphological Changes in Mitochondria under Salt Stress*

Mitochondria play a key role in plant tolerance to salt stress [142]. It has been proved that the impairment of mitochondrial function can lead to the hypersensitivity of plants to salt stress. For example, the mutation of Arabidopsis RNA editing factor *SLO2* affects the mitochondrial electron transport chain, resulting in a large amount of ROS production, and the root of Arabidopsis seedlings was damaged on the salt stress medium [143]. Plant mitochondria respond to PCD signals under severe salt stress by undergoing a permeability transition, releasing cytochrome c, and decreasing ATP production, ROS outbreak, and mitochondrial morphological changes [144,145]. Mitochondria need to constantly regulate their functions to maintain energy homeostasis and reduce the damage caused by such oxidative stress [146].

The factors that influence mitochondrial functions can change mitochondrial morphology under salt stress. The pH value in mitochondria determines the rate of oxidative phosphorylation [147] and affects mitochondrial ATP synthesis and ROS levels. Under exogenous NaCl treatment, ROS derived from NADPH oxidase promoted mitochondrial alkalization and caused mitochondria to become aggregated under salt stress [148]. It has been reported that plant mitochondria can respond to salt stress by forming a mitochondrial network [149] through the bulge of OM or by direct contact with each other. This kind of network may prevent the rupture of OM caused by salt stress and ensure the survival of cells, and the increase in matrix volume may also increase the activity of the respiratory chain and the production of ATP.

Mitochondria move along actin microfilaments (MFs) in plants, which are essential to mitochondrial arrangement, distribution, and function. In *Arabidopsis thaliana*, the actin-related Protein2/3 (ARP2/3) complex regulates mitochondrial-associated calcium signaling during salt stress [150]. Plants lacking ARP2 increased the content of salt-induced cytosolic $Ca^{2+}$, decreased mitochondria movement, and made mitochondria aggregated. Additionally, the mitochondrial permeability transition pore opened, and the mitochondrial membrane potential Ψm was impaired in *arp2* mutant. Those changes were associated with salt-induced PCD. In another study, it was found that accumulation of the cytoplasmic male sterility (CMS) protein ORFH79 resulted in the dysfunction of mitochondria with decreased

enzymatic activities of respiratory chain complexes, reduced the level of ATP, and even changed mitochondrial morphology [151]. However, overexpressing a fertility restorer gene *Rf5* inhibited mitochondrial dysfunction caused by ORFH79, restored mitochondrial morphology, and improved plant tolerance to drought and salt stress [151]. The mitochondrial transcription termination factor (mTERF) protein family affects gene expression in plastid and mitochondrial genomes [152]. The transcript levels of some mitochondrion-encoded genes were reduced in *mterf27* mutant. Importantly, the loss of mTERF27 function led to developmental defects in mitochondria under salt stress. The morphology was manifested as mitochondrial vacuolation and irregular cristae structure [153].

In conclusion, the changes in mitochondrial morphology under salt stress cannot be simply attributed to osmotic stress. The stress responses of plant mitochondria in the face of salt stress will affect the mitochondrial dynamics process. Meanwhile, the changes in mitochondrial morphology will provide a favorable membrane and substrate environment for the play of mitochondrial functions.

### 5.4. Morphological Changes in Mitochondria under Other Abiotic Stress

Among abiotic stress faced by plants, temperature, drought, and salt stress are the main environmental factors that affect the geographical distribution of plants, limit crop yield, and threaten food security. However, with the increasingly frequent occurrence of extreme weather, the damage caused by ozone, heavy metal poisoning in soil, hypoxia stress to plant growth and development, and the adverse impact on agricultural production should not be underestimated.

Morphological changes in plant mitochondria under this stress are also noteworthy. Recently, particularly promising studies have focused on the changes in the size and number of mitochondria in several plants under different stress conditions (Table 1).

**Table 1.** Mitochondrial morphological changes under several typical abiotic stress.

| Type of Stress | Species | Parts | Morphology of Mitochondria | | Reference |
|---|---|---|---|---|---|
| | | | **Size** | **Amount** | |
| Low oxygen pressure | *Nicotiana tabacum* L. | Mesophyll cells | Giant mitochondria; eventually became an extensive mitochondrial reticulum, including large plates | The number of mitochondria decreased | [154] |
| | *Arabidopsis thaliana* L. | Leaf cells | Large and elongated | – | [155] |
| Ozone stress | *Fagus sylvatica* L. | Beech foliage cells | Degeneration of cristae and matrix in mitochondria | – | [156] |
| | *Picea abies* L. | Mesophyll cells | The size of mitochondria decreased | Numerous | [157] |
| UV stress | *Palmaria palmata* L. | Algal cells | Cristae were visible and even appeared swollen | – | [158] |
| | *Arabidopsis thaliana* L. | Leaf cells | The mitochondria clustered irregularly surrounding the chloroplasts or elsewhere within the cytoplasm | – | [159] |
| | *Arabidopsis thaliana* L. | Leaf cells (*agt* mutant) | Small and fragmented | Numerous | [89] |
| | *Arabidopsis thaliana* L. | Leaf cells (wild type) | – | The number of mitochondria decreases | [89] |

**Table 1.** *Cont.*

| Type of Stress | Species | Parts | Morphology of Mitochondria | | Reference |
|---|---|---|---|---|---|
| | | | Size | Amount | |
| Acid rain | *Lycopersicon esculentum* M. | Leaf cells | Swollen, vacuolated, and cristae collapsed | – | [160] |
| Silver nanoparticles (AgNPs) | *Hordeum vulgare* L. | Leaf cells | The mitochondrial cristae were partially or totally degenerated | – | [161] |
| Methyl jasmonate, MeJa | *Arabidopsis thaliana* L. | Leaf cells | Swollen and spherical | – | [162] |

The relationship between the size and number of mitochondria under stress is very subtle. The appearance of a large mitochondrial network is accompanied by the decrease in the number of mitochondria [156], whereas the fragmentation of mitochondria is accompanied by an increase in the number of thread particles [89]. This implies that plant mitochondria may respond to mild stress by fusing into a larger mitochondrial network or by dividing to increase the number in cells, thus providing a sufficient energy supply to cells (Figure 2). However, when stress persists or deepens, the fragmentation and collapse of cristae leads to disintegration of the mitochondrial structure, which implies the loss of mitochondrial functions. Plant cells need to remove these mitochondria through the autophagy pathway to prevent the accumulation of toxic substances.

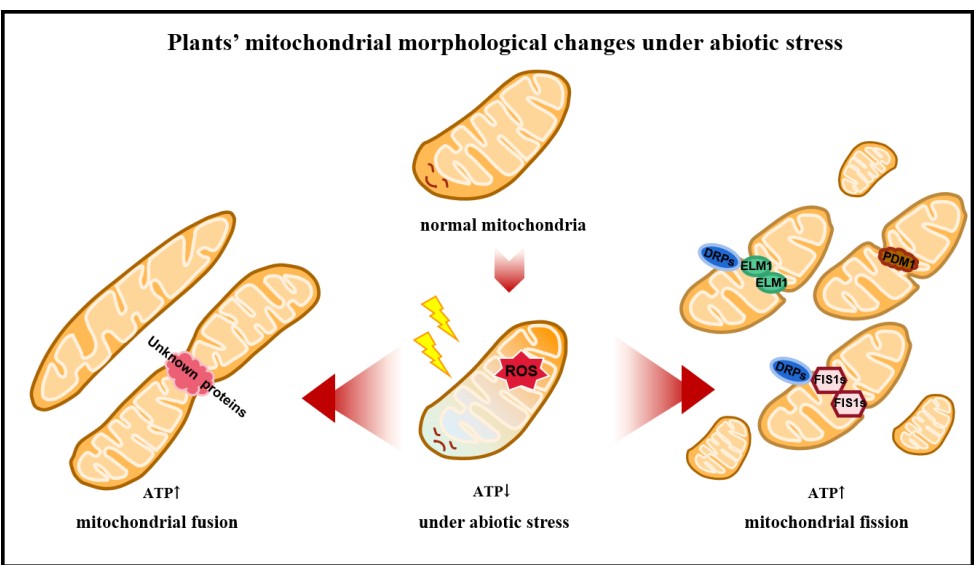

**Figure 2.** Morphological changes in plant mitochondria under abiotic stress. Image credit: Hui Tang.

## 6. Conclusions

Plant mitochondria serve as important organelles responding to environmental tension, whose fine structure has a significant impact on homeostasis in plant cells. Morphological changes in the face of abiotic stress may be the first response after mitochondria integrate stress signals. This microscopic morphological and dynamic changes will directly affect the homeostasis of the mitochondrial environment and induce PCD events, such as an excessive release of ROS.

We have discussed the changes in the ultrastructure of plant mitochondria in the face of abiotic stress and proposed a potential relationship between abiotic stress and plant mitochondrial fusion–fission–autophagy. However, we acknowledge that our study has not been explained in detail at the mechanistic level, and it is still unclear how to represent the cascade of mitochondrial morphology and dynamics with biochemical levels, such as

ROS and oxidative phosphorylation under stress conditions, as well as the response to the abundance and stability of related proteins. In the future, the study of mitochondrial metabolomics and proteomics is expected to provide new data. Linking morphological changes in plant mitochondria to the extreme climate stress that plants are facing helps to understand the molecular nature of organelle stress responses, which is crucial for plant genetic breeding under the dramatic change in the global natural environment.

**Author Contributions:** H.T. completed the literature review and writing of this review. H.Z. reviewed the review and directed the revisions. All authors have read and agreed to the published version of the manuscript.

**Funding:** This review was supported by the Joint NSFC-ISF Research Program (Grant No. 32061143022) and the 2115 Talent Development Program of China Agricultural University (Grant No. 1061-00109017) to HZ.

**Data Availability Statement:** Not applicable.

**Conflicts of Interest:** The authors declare no conflict of interest.

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
