# Peer review of "Specific Changes in Morphology and Dynamics of Plant Mitochondria under Abiotic Stress"

_horticulturae, doi:10.3390/horticulturae9010011_

Round 1
Reviewer 1 Report
Title of the paper:
The paper's title is an enormous scope of the changes in morphology and dynamics of plant mitochondria under abiotic stress. My suggestion to authors is to write about specific changes in morphology and dynamics of plant mitochondria under drought stress or salt stress to be more specific and to have a high contribution to the reader.
1: Introduction:
The introduction section is relatively short and very good; however, there are some spaces for authors to enhance its quality further, which are as follows:
1. The authors should insert specific information about mitochondria of higher plants, such as the definition, which is vital as a site of oxidative energy metabolism and synthesise the ATP in plants.
2. The authors should insert the mechanisms and molecules involved in plant mitochondrial dynamics.
2: Morphological and dynamic changes of mitochondria
1. Lines 84-86: Figure 1. Please mention the references for Figure 1.
4. Morphological changes of mitochondria in response to abiotic stresses
The authors discussed the morphological changes of mitochondria under temperature, drought, salt and under other abiotic stress.
The information was general, and I suggest selecting one of abiotic stress and discussing and explaining the deepest.
Lines 477: Figure 2: Please insert the reference for this figure.
5. Conclusion
The conclusion is very long and should be inserted into the main findings.
Author Response
Dear Editor and Reviewers,
We would like to express our sincere appreciation for the time and effort spent by the two reviewers in the evaluation of our manuscript titled “Specific changes in morphology and dynamics of plant mitochondria under abiotic stress” (horticulturae-2092450). Thank you very much for your positive and constructive comments and suggestions, which are very helpful for the improvement of our paper. We have considered the comments carefully and have revised the manuscript thoroughly based on the comments. We deeply appreciate your work, and hope that the corrections and responses will meet with your approval. Revised portions are marked, in different colours depending on the reviewers, in the revised manuscript and the point to point responses to the comments are listed below in this cover letter. We look forward to your information about our revised paper.
Best regards,
Yours sincerely,
Dr. Zhu Hongliang
REVIEWER 1 (BLUE)
Title of the paper:
The paper's title is an enormous scope of the changes in morphology and dynamics of plant mitochondria under abiotic stress. My suggestion to authors is to write about specific changes in morphology and dynamics of plant mitochondria under drought stress or salt stress to be more specific and to have a high contribution to the reader.
- Special thanks for your contribution. We have changed the title in accordance with the reviewer's instructions. Lines 2-3.
1: Introduction:
The introduction section is relatively short and very good; however, there are some spaces for authors to enhance its quality further, which are as follows:
1. The authors should insert specific information about mitochondria of higher plants, such as the definition, which is vital as a site of oxidative energy metabolism and synthesise the ATP in plants.
2. The authors should insert the mechanisms and molecules involved in plant mitochondrial dynamics.
- Special thanks for your comment. We have modified the introduction section to supplement the definition of higher plant mitochondria and the mechanisms of mitochondrial dynamics in the manuscript according to the reviewer's recommendation. Lines 26-42.
2. Morphological and dynamic changes of mitochondria
1. Lines 84-86: Figure 1. Please mention the references for Figure 1.
- Thank you very much for your advice. We drew Figure 1 independently according to the previous research results, and the relevant mechanism and research results were shown in “2.1 Ultrastructure of mitochondria” section. References 18-23.
4. Morphological changes of mitochondria in response to abiotic stresses
The authors discussed the morphological changes of mitochondria under temperature, drought, salt and under other abiotic stress.
The information was general, and I suggest selecting one of abiotic stress and discussing and explaining the deepest.
- Special thanks for your careful guidance. We have modified and refined the section of “5. Morphological changes of mitochondria in response to abiotic stress” based on your comments. And the section of “5.2 Morphological changes of mitochondria under drought stress” has been discussed in depth.
Lines 477: Figure 2: Please insert the reference for this figure.
- Thank you very much. We summarized the previous researches results and drew Figure 2 by ourselves. The descriptions of the fusion and fission related proteins of mitochondria in the text have been introduced into references.
5. Conclusion
The conclusion is very long and should be inserted into the main findings.
- Thank you very much for your observation. We have refined the language in the conclusion section. Lines 504-509. In addition, we have supplemented the text with our mean findings. Lines 488-491.

Reviewer 2 Report
The article has a certain theoretical value. However, I have some minor edits.
Please write the authors of Figures 1 and 2. In the text, the authors should write in more detail how the study of the structure of chloroplasts (anatomy, morphology) will explain the issues of genetic selection and adaptive selection. Possibly this will help to understand some issues in the introduction of plants.
Author Response
Dear Editor and Reviewers,
We would like to express our sincere appreciation for the time and effort spent by the two reviewers in the evaluation of our manuscript titled “Specific changes in morphology and dynamics of plant mitochondria under abiotic stress” (horticulturae-2092450). Thank you very much for your positive and constructive comments and suggestions, which are very helpful for the improvement of our paper. We have considered the comments carefully and have revised the manuscript thoroughly based on the comments. We deeply appreciate your work, and hope that the corrections and responses will meet with your approval. Revised portions are marked, in different colours depending on the reviewers, in the revised manuscript and the point to point responses to the comments are listed below in this cover letter. We look forward to your information about our revised paper.
Best regards,
Yours sincerely,
Dr. Zhu Hongliang
REVIEWER 2 (GREEN)
The article has a certain theoretical value. However, I have some minor edits.
Please write the authors of Figures 1 and 2. In the text, the authors should write in more detail how the study of the structure of chloroplasts (anatomy, morphology) will explain the issues of genetic selection and adaptive selection. Possibly this will help to understand some issues in the introduction of plants.
- Thank you very much for your careful guidance.
1.According to the existing reports on mitochondrial microstructure and morphology related proteins, we have drawn Figure 1 and Figure 2 by ourselves. The descriptions of mitochondrial microstructure as well as fusion and fission related proteins in the text have been introduced into references.
The main objective of this manuscript was to analysis the morphological and dynamics changes of mitochondria under abiotic stress in higher plants. The morphology and structure of chloroplasts are beyond our consideration. However, we have added a section focusing on the effects of interactions between mitochondria and other organelles (including endoplasmic reticulum, chloroplasts, and peroxisomes) on morphological and dynamics processes in higher plants, which may help us understand the role of the network of connections between organelles on the introduction of plants. Lines 276-301. Nevertheless, it is a good suggestion that we will take into account and apply in further studies.

Round 2
Reviewer 1 Report
The authors improved and revised the paper according to my comments.
Author Response
Thanks for reviewing our manuscript.